# Angiogenic Events Positively Modulated by Complex Magnetic Fields in an In Vitro Endothelial Cell Model

**DOI:** 10.3390/cells14050332

**Published:** 2025-02-24

**Authors:** Alessia Ricci, Amelia Cataldi, Marialucia Gallorini, Viviana di Giacomo, Monica Rapino, Natalia Di Pietro, Marco Mantarro, Adriano Piattelli, Susi Zara

**Affiliations:** 1Department of Pharmacy, “G. d’Annunzio” University of Chieti-Pescara, 66100 Chieti, Italy; amelia.cataldi@unich.it (A.C.); marialucia.gallorini@unich.it (M.G.); viviana.digiacomo@unich.it (V.d.G.); susi.zara@unich.it (S.Z.); 2Ud’A Techlab, “G. d’Annunzio” University of Chieti-Pescara, 66100 Chieti, Italy; 3Unit of Chieti, Genetic Molecular Institute of CNR, “G. d’Annunzio” University of Chieti-Pescara, 66100 Chieti, Italy; m.rapino@unich.it; 4Department of Medical, Oral and Biotechnological Sciences, “G. d’Annunzio” University of Chieti-Pescara, 66100 Chieti, Italy; natalia.dipietro@unich.it; 5Center for Advanced Studies and Technologies (CAST), “G. d’Annunzio” University of Chieti-Pescara, 66100 Chieti, Italy; 6Independent Researcher, MFI Srl, 00163 Rome, Italy; marco.mantarro@mfisrl.com; 7School of Dentistry, UniCamillus-Saint Camillus International University of Health and Medical Sciences, 00131 Rome, Italy; apiattelli51@gmail.com

**Keywords:** endothelial cells, angiogenesis, complex magnetic fields, extracellular matrix remodeling, tissue regeneration, healing processes

## Abstract

The vascular system is primarily responsible for orchestrating the underlying healing processes to achieve tissue regeneration, thus the promotion of angiogenic events could be a useful strategy to repair injured tissues. Among several approaches to stimulate tissue regeneration, non-invasive devices are currently widely diffused. Complex Magnetic Fields (CMFs) are innovative pulsed multifrequency electromagnetic fields used for their promising results in clinical applications, such as diabetic foot treatment or edema resorption. Nevertheless, few papers are available demonstrating the biological mechanisms involved. In this paper, in order to understand CMFs’ capability to promote angiogenic events, Regenerative Tissue Program (RTP) was applied to an in vitro Endothelial Cells (ECs) model. ECs were stimulated with (I) 2 RTP consecutive cycles, (II) with an interval of 8 h (T0 + T8), or (III) 24 h (T0 + T24) from one cycle to another. Results demonstrate that (I) extracellular matrix degradation is promoted through matrix metalloproteinases 2 and 9 modulation, leading to an increased cell migratory capability; (II) CMFs support EC growth, activating Integrin β1-Erk-Cdk2 pathway and sustaining G1/S transition; (III) vessel morphogenesis is promoted when CMFs are applied. In conclusion, the promising clinical results are supported by in vitro analyses which evidence that main angiogenic events are stimulated by CMFs.

## 1. Introduction

One of the main intriguing topics for the scientific community is to promote tissue regeneration after damage, inflammation, oxidative stress, and stressful stimulations [1]. During the healing processes of damaged tissues, one of the main roles is played by the vascular system: the vascular niche formed by blood vessels and surrounding cells is essential for both local and distant signaling, which in turn shapes the regenerative response [2]. Indeed, in general, the angiogenic process is crucial in supporting the growth and the health of all organs through oxygen and nutrient supply and waste elimination; vascular system dysfunction is the principal cause of human morbidity and mortality [3]. Particularly, during the healing processes of damaged tissues, vasculogenic mediators, produced as a result of local hypoxia and inflammation upon tissue damage, coordinate the migration of precursor Endothelial Cells (ECs) from their niche environment to the site of damage [4]. The development of a new vessel network, namely, angiogenesis, requires 4 main steps: Extracellular Matrix (ECM) degradation through the activity of Matrix Metalloproteinases (MMPs), migration, proliferation, and morphogenesis [5]. The mature ECs, through the release of ECM proteins, determine ECM deposition that favors the stabilization of new blood vessels, which, in turn, supply oxygen and nutrients to tissue and organs [4,6]. One of the main endothelium-derived molecules, able to orchestrate the angiogenic steps, is Nitric Oxide (NO). NO is a small signaling molecule that can be produced in all cell types, thanks to the activity of three different Nitric Oxide Synthase (NOS) enzymes: inducible NOS (iNOS), neuronal NOS (nNOS), and eNOS [7]. iNOS is the inducible form of NOS, mainly activated during inflammation or pathophysiologic events [8]; nNOS is a specific isoform expressed in central and peripheral neurons [9] while eNOS is the isoform selectively expressed in ECs [10]. NO, released both from differentiated and from pro-genitor ECs, is an essential cell mediator that preserves the normal activity of ECs functioning as regulators of different angiogenic events. Indeed, it stimulates progenitor and mature EC migration, it regulates EC proliferation and survival when stressful or inflamed stimulations occur, it is essential for the morphogenic event and, lastly, after blood vessels formation and stabilization, NO release controls vasodilatation and inhibits platelets aggregation [11]. Several post-translational modifications are required to activate or inhibit eNOS activity, such as phosphorylation and acetylation.

Integrin β1 is an essential glycoprotein receptor with a context-dependent function in ECs [12]. It is known that activation of integrins can provoke Erk activation and its translocation into the nucleus where, acting on other mediators such as Elk1, influences the activity of cyclins and cyclin-dependent kinases, with a positive modulation of the cell cycle [13]. In particular, integrin-mediated cell adhesion is essential for G1/S checkpoint progression and Cdk2/cyclin E is one of the cyclin complexes indirectly modulated by Integrin-Erk activation, sustaining G1 to S transition.

With the aim of finding new systems able to promote tissue regeneration, healing or managing pain resulting from tissue injury, research of non-invasive technologies is widely diffused. Devices committed to produce Magnetic Fields (MFs) at different intensities and frequencies are scientifically validated tools in modern medicine, widely used to effectively treat inflammation and oxidative stress often associated with several diseases, such as osteoarthritis, neuropathy pain, and others [14,15,16,17]. MFs act by opening and gating membrane channels and inducing alterations in ion concentration, stimulating cell interactions, cell movement, cytoskeleton rearrangement, modulation of cell growth and differentiation, wound healing, and anti-inflammatory response, along with having an antiproliferative impact on cancer cells [18]. Vascular system stimulation with MFs, in order to maintain a physiological state, has been already assessed since ages [19,20]. In fact, it is commonly known that MFs increase tissue blood flow and speed up body metabolism. MFs have a significant impact on the relaxation and constriction of smooth muscles of capillary vessels, modifying blood flow: rats exposed to 70 mT MFs during in vivo tests showed a noticeable rise in blood flow as a result of blood vessel dilatation [21]. In the area of new non-invasive tools emitting MFs, a novel device able to produce Complex Magnetic Fields (CMFs) has been developed and applied to counteract different pathological conditions. The aforementioned electronic device emits pulsed multifrequency electromagnetic fields ranging from 1 to 250 µT characterized by variable intensity, frequency, complex wave form, and time stimulation. While several in vitro studies highlighted the molecular mechanisms activated by MFs on ECs [22,23], little is known about the biological processes mainly promoted or suppressed by CMFs and which molecular pathways are recruited to get specific biological responses [24,25]. In particular, since the in vivo effects of CMF application are already known, the aims of the present study are (I) to set up, for the first time, a biological in vitro model in which CMFs are applied to ECs, represented by EA.hy926 cell lineage; (II) to understand the effect of CMFs on the angiogenic events in a physiological condition, and (III) to determine how the four angiogenic steps are modulated by CMFs, focusing on the possible molecular pathways recruited by this system.

## 2. Materials and Methods

### 2.1. Cell Culture

The human umbilical vein cell line EA.hy926 was purchased from ATCC (LGC Standards S.r.l., Milan, Italy) and cultured in DMEM supplemented with 10% Fetal Bovine Serum (FBS), 1% penicillin/streptomycin, and 4 mM L-glutamine (all from Euroclone S.p.A., Milan, Italy), at 37 °C with 5% CO_2_.

### 2.2. Cell Stimulation

The electronic device emitting CMFs, Next sx version (M.F.I. Medicina Fisica Integrata, Rome, Italy), is able to provide pulsed multifrequency electromagnetic fields ranging from 1 to 250 µT with different features of intensity, frequency, complex wave form, and time stimulation (Figure 1).

The dimensions of the CMF generator are: width 170 mm, length 170 mm, height 50 mm, whereas the dimensions of the emitting plaque are: length 360 mm, width 120 mm, height 25 mm.

The magnetic field is delivered by three separated solenoids, spaced 1 cm, that deliver the same field. The solenoids are made by winding 650 turns of 0.35 mm wide enameled copper wire. The coil’s external dimensions are 110 mm, internal 12 mm, and the coil’s thickness is 8 mm. Magnetic field lines are applied to the sample being treated at a 90° angle. Six 35 mm Petri dishes or three culture flasks of 25 cm^2^ or two multiwell plates can be placed on the CMF emitting plate at once.

The CMF generator includes several programs working in relation to the sector of the application. Each program has different steps and each of them is characterized by different intensities (1–250 µT), frequencies (1–250 Hz), interval times (1–4 min each step), and forms of the complex multifrequency waves with harmonic enrichments. Frequency, induction of intensity, wave form, and time stimulation represent one of the steps of the machine program. The final program is generally generated by 6 to 10 steps. The information concerning the programs of the device is patent pending. In this experimental protocol, the parameters used were: frequency from 1 to 112 Hz, intensity of the induction from 1 to 195 µT, duration time of steps from 1 to 4 min/each, and impulsive waveforms with odd multiple harmonics.

Regenerative Tissue Program (RTP) with a duration of 28 min was applied. EA.hy926 underwent a CMF stimulation repeating the RTP program at different time intervals, as reported in Table 1.

Cells seeded in multiwell-plates were placed on the CMF emitting plaque (Figure 1B) and located in the incubator for all the duration of the stimulation. Controls (UT1 and UT2) were arranged in the incubator without receiving stimulation. The UT1 represents the control for the first treatments block, in which a short interval time elapses between the repetition of RTP cycles. The UT2 is the control for the second treatments block, in which the interval time between the repetition of RTP cycles is 24 h, thus, UT2 remains in culture 24 h more than UT1. After an in-depth evaluation of cell viability testing all the conditions listed in Table 1, the best RTP repetitions, selected for further experiments, were 2 cycles, T0 + T8 and T0 + T24 (Figure 2). At the established experimental times, cells were collected for subsequent analyses.

### 2.3. Cell Metabolic Activity (MTT) Assay

EA.hy926 were seeded in 35 mm Petri dishes having an area of 8 cm^2^, with a cell density of 170,000 cells/dish. After the exposure to the aforementioned experimental conditions of CMFs, cells were subjected to a metabolic activity test (MTT). The MTT (3-[4,5-dimethylthiazol-2-yl]-2,5 diphenyl tetrazolium bromide) test evaluates the capability of viable cells to convert MTT into violet formazan salts which are then dissolved by DMSO. After 24 and 48 h, the culture medium was removed and replaced by a fresh one containing 10% of MTT (Merck Life Science, Milan, Italy), incubated for 4 h and then in DMSO for 30 min at 37 °C. Absorbance was spectrophotometrically read by a microplate reader at 540 nm wavelength (Multiskan GO, Thermo Scientific, Waltham, MA, USA). The obtained values were normalized with untreated cell values (UT1 or UT2) and expressed as percentage of cell viability, as previously reported [26].

### 2.4. Lactate Dehydrogenase (LDH) Assay

The amount of LDH release in the culture media was quantified by means of the CytoTox 96 Non-Radioactive Assay (Promega Corporation, Fitchburg, WI, USA) to evaluate the cytotoxic effect exerted by CMFs on EA.hy926 cells. Supernatants and LDH reaction mixture were pipetted in a 96-well plate with a flat bottom (Falcon, Corning Incorporated, New York, NY, USA) in 1:1 proportion and incubated for 30 min in the dark. After that, 50 μL of stop solution was added and the absorbance was read at 490 and 690 nm wavelength by means of Multiscan GO microplate reader (Thermo Scientific, Waltham, MA, USA). Released LDH was normalized on MTT and then a fold increase on the corresponding UT condition was calculated.

### 2.5. Phase Contrast Images

Images of the different experimental conditions were acquired in phase contrast by an inverted light microscope Leica DMI1 (Leica Cambridge Ltd., Cambridge, UK Leica) with a Leica MC120 HD camera (Leica Cambridge Ltd., Cambridge, UK).

### 2.6. Trypan Blue Dye Exclusion Test

Cell viability was assessed by Trypan blue test. After the specific stimulations, cells were washed with PBS, trypsinized, and processed for Trypan blue dye exclusion test, which selectively discriminates live cells from the blue-coloured dead cells, counted in a Burker chamber. The number of live cells is reported as number of cells/mL of solution.

### 2.7. Protein Extraction and Western Blotting Analysis

EA.hy926 cells were seeded in a 6-well plate at 200,000 cells/well, and 14 h and 24 h after the specific CMF stimulation, cells were trypsinized, centrifuged, and collected in cold PBS. Pellets were then lysed to obtain a pool of proteins, quantified by bicinchoninic acid assay (QuantiPro™ BCA Assay kit, Merck Life Science, Milan, Italy) [27]; 25 µg of lysate were run on polyacrylamide gel by electrophoresis and transferred to nitrocellulose membranes. The membranes were probed overnight with mouse monoclonal anti-tubulin (dilution 1:5000), anti-MMP-1, anti-MMP-2, anti-MMP-9 antibodies (all diluted 1:200), rabbit polyclonal anti-Cyclin dependent kinase 2 (Cdk2), anti-endothelial Nitric Oxide Synthetase (eNOS) antibodies (both diluted 1:200) (all from Santa Cruz Biotechnology, Santa Cruz, CA, USA), rabbit monoclonal anti-Integrin β1, anti-Erk and anti-p-Erk, anti-p-eNOS (ser1177), anti-p-eNOS (thr495) (all diluted 1:1000) (all from Cell Signaling Technology, Danvers, MA, USA). The day after the incubation, membranes were probed in the presence of specific IgG horseradish peroxidase (HRP)- conjugated secondary antibodies (Calbiochem, Darmstadt, Germany). The ECL detection system (LiteAblot Extend Chemiluminescent Substrate, EuroClone S.p.a., Milan, Italy) was used to reveal immunoreactive bands, and densitometry was performed by means of a ChemiDocTM XRS system and a QuantiOne 1-D analysis software, version 4.6.6 (BIORAD, Richmond, CA, USA), reported as Integrated Optical Intensity. The densitometric values of internal tubulin were used to standardize the obtained data.

### 2.8. RNA Extraction, Reverse Transcription (RT), and Real-Time RT-Polymerase Chain Reaction (RT-PCR)

EA.hy926 cells were seeded in 6-well plates at 200,000 cells/well; 6 and 14 h after the specific stimulations, EA.hy926 cells were detached by trypsin and harvested after centrifugation. RNA PureLink^®^ RNA Mini Kit (Life Technologies, Carlsbad, CA, USA) was used for RNA extraction, following the manufacturer’s instructions. To remove DNA contamination, samples were incubated for 15 min in the presence of 80 µL of DNase mixture (On-column PureLink^®^ DNase Treatment, Life Technologies, Carlsbad, CA, USA). Extracted RNA was eluted in 30 µL of Nuclease-Free Water and RNA concentration (ng/µL) was assessed by Qubit^®^ RNA BR Assay Kits (Life Technologies, Carlsbad, CA, USA). Then, 400 ng of RNA were reverse transcribed by using a High-capacity cDNA Reverse Transcription Kit (Life Technologies, Carlsbad, CA, USA). Gene expression was analyzed for all the examined mRNAs by quantitative PCR using PowerUp^TM^ SYBR^TM^ Green Master Mix (2×) (Thermo Fisher Scientific, Waltham, MA, USA). Each reaction of amplification was carried out in a MicroAmp^®^ Optical 96-well Reaction Plate (Life Technologies, Carlsbad, CA, USA) by adding 10 µL of SYBR Green, 1 µM of forward and reverse primers (stock solution 100 µM), and 10 ng of cDNA and Nuclease-Free Water to reach 20 µL of final volume. Primer sequences used are reported in Table 2.

The run method suggested by the datasheet was performed in QuantStudio 3 (Thermo Fisher Scientific, Waltham, MA, USA). Gene expression data were analyzed by using QuantStudio™ Design & Analysis Software v1.5.1 (Thermo Fisher Scientific, Waltham, MA, USA). The gene expression data were normalized to the expression level of *GAPDH*. To quantify the relative abundance of mRNA, the comparative 2^−ΔΔCt^ method was applied (relative quantification).

### 2.9. Wound Healing Assay

EA.hy926 cells were seeded in 6-well plates at 250,000 cells/well. After cells reached 70–80% of cell confluence in each well, the cell monolayer was scratched with a p200 pipet tip (Falcon, Corning Incorporated, New York, NY, USA). After having been washed with PBS to remove debris, cells were stimulated with the specific CMFs treatment. Images were acquired after 0, 14, 24, and 48 h of stimulation with an inverted light microscope Leica DMI1 (Leica Cambridge Ltd., Cambridge, UK) with a Leica MC120 HD camera (Leica Cambridge Ltd., Cambridge, UK) to obtain computerized images.

### 2.10. Cell Cycle Analysis

In the cell cycle analysis, 160,000 cells/well of a 6-well plate were seeded and 38 h after 2 cycles and T0 + T8 stimulations were fixed overnight with cold ethanol 70% *v*/*v*. The day after, samples were incubated in the presence of a staining solution containing PBS (300 µL), Rnase (100 µg/mL, stock solution 10 mg/mL in 10 mM sodium acetate buffer, pH 7.40), and propidium iodide (PI) (10 µg/mL, stock solution 1 mg/mL in water) (all from Sigma Aldrich, MI, USA) and kept overnight at 4 °C in the dark. The PI fluorescence was revealed by a flow cytometer with a 488 nm laser (CytoFlex flow cytometer, Beckman Coulter, Brea, CA, USA) in the FL-3 channel; 15,000 events/sample were collected and analyzed by CytExpert Software version 5.0 (Beckman Coulter, Brea, CA, USA). Cells in the G1, S, or G2 phase of the cell cycle, expressed as percentages, were calculated through the ModFit LT™ Software, version 5.0 (Verity Software House, Topsham, ME, USA).

### 2.11. Tube Formation Assay

Tube formation assay was performed as previously described [28]. Briefly, 50 µL of ECM Gel (8–12 mg/mL, Merck Life Science, Milan, Italy) was dropped onto each well of a 96-well plate and were allowed to solidify for 1 h at 37 °C in a humidified 5% CO_2_ incubator; 3 × 10^4^ cells were seeded in serum-free medium onto the ECM-coated wells. The cells were left to stabilize onto the ECM for 30 min and then the specific CMFs stimulation was applied; 8 h and 20 h after the stimulations, computerized images were acquired by means of an inverted light microscope Leica DMI1 (Leica Cambridge Ltd., Cambridge, UK) with a Leica MC120 HD camera (Leica Cambridge Ltd., Cambridge, UK). Tubule-like structures were quantified in terms of number of meshes, segments, extremity, junctions, and nodes by using ImageJ software, version 1.54 k, and Angiogenesis Analyzer plugin.

### 2.12. Intracellular Nitric Oxide (NO) Production

A Nitrite Assay Kit (Griess Reagent) (Merck Life Science, Milan, Italy) was used to determine intracellular NO production, following the manufacturer’s instructions. Briefly, after CMFs stimulation, 1 × 10^6^ cells were homogenized with ice-cold Nitrite Assay Buffer and chilled on ice for 10 min. Then, samples were centrifuged at 10,000× *g* at 4 °C for 5 min and the supernatants were collected, loaded in a 96-well plate with a Griess Reagents mixture in a 1:1 proportion and left to incubate for 10 min. Absorbance was spectrophotometrically read by a microplate reader at 540 nm wavelength (Multiskan GO, Thermo Scientific, Waltham, MA, USA) and the obtained values were interpolated with a Nitrite standard curve.

### 2.13. Statistical Analysis

Data were statistically analyzed by GraphPad 9 software (San Diego, CA, USA) using ordinary one-way ANOVA followed by post hoc Tukey’s multiple comparison test by means of the Prism 5.0 software (Graph-Pad, San Diego, CA, USA). The results are the mean values ± SD. Values of *p* ≤ 0.05 were considered statistically significant.

## 3. Results

### 3.1. Evaluation of CMF System Biocompatibility on EA.hy926 ECs

The effect of CMF RTP on EA.hy926 was evaluated by MTT assay, which was performed 24 and 48 h after stimulation to assess the biocompatibility of the system; 24 h after 2 and 3 RTP consecutive cycles, T0 + T4 and T0 + T8, samples show a statistically significant increase in metabolic activity compared to U1; 1 cycle and T0 + T4 + T8 samples appear similar to UT1 with no significant changes. The same situation is recorded when RTP cycles are repeated after 24 h: T0 + T24 and 2T0 + 2T24 show a similar cell metabolic activity compared to UT2 (Figure 3A); 48 h after RTP stimulations, no statistically significant changes among treated conditions and UT1 or UT2 are detectable, except for a significant reduction in cell metabolic activity after 1 cycle stimulation compared to UT1 and a slight increase in cell metabolic activity after 2T0 + 2T24 RTP cycles compared to UT2 (Figure 3B). Based on increased metabolic activity observed after 24 h, 3 different conditions were selected for further experiments: 2 consecutive cycles, T0 + T8 and T0 + T24.

To exclude a possible cytotoxicity of the selected RTP conditions, LDH release assay, 14 and 24 h after the stimulation, was performed. Compared to both UT samples, no statistically significant changes are evident in LDH release after both time points (Figure 4).

In parallel, a morphological analysis, performed by light phase contrast microscopy observation, evidences that 24 h after CMF stimulation, cells appear similar to the non-stimulated one. However, an irregular distribution of cells in UT1 and UT2 is noticed, when compared to CMF stimulated cells, the latter appearing more confluent (Figure 5A). This observation is corroborated by Trypan blue exclusion test which evidences a statistically significant increase in treated cell number, compared to both UT1 and UT2 (Figure 5B); 48 h after the specific stimulation, in T0 + T8 condition, a statistically significant increase in cell number, compared to UT1, is disclosed (Figure 5C).

### 3.2. ECs Stimulation by CMFs Regulates eNOS Phosphorylation and NO Production

A crucial role in the correct physiology of the vascular system is the balance between the activated form of eNOS (phosphorylated on ser1177) and the non-activated form of eNOS (phosphorylated on thr495), with a regulated production of NO. With the aim of evaluating the effect of CMFs on eNOS activity, gene and protein expression of eNOS, along with an evaluation of NO production were investigated.

Six hours after the stimulations, only T0 + T24 condition discloses a statistically significant increase in *eNOS* gene expression compared to UT2 (Figure 6A). After 14 h, T0 + T8 significantly increases *eNOS* gene expression with respect to UT1; T0 + T24 stimulation maintains increased levels of *eNOS* compared to UT2, recording the same trend of 6 h (Figure 6B).

p-eNOS (ser1177)/eNOS ratio increases in a statistically significant manner 14 h after T0 + T24 stimulation compared to UT2, while in 2 cycles and T0 + T8 samples similar levels of UT1 are shown (Figure 6C). However, after 24 h the ratio between the total eNOS and the ser-1177 phosphorylated eNOS significantly increases when ECs are stimulated with T0 + T8 RTP compared to UT1, while no changes are recorded in the other experimental points (Figure 6D).

p-eNOS (thr495)/eNOS ratio shows an opposite trend compared to the phosphorylation on ser1177: 14 h after the RTP stimuli, no significant changes are recorded respect to UT1 and UT2 (Figure 6E). However, after 24 h an increased e-NOS (thr495) phosphorylation is evident in UT1, with a significant reduction in p-eNOS (thr495)/eNOS ratio after 2 cycles and T0 + T8 stimulations (Figure 6F). The measurement of intracellular NO levels seems to reflect the p-eNOS (ser1177) protein levels: after 14 h, 2 cycles and T0 + T8 show the same NO levels production of UT1. A strong and significant increase is detected after T0 + T24 stimulation with respect to UT2 (Figure 7A). After 24 h, 2 cycles slightly but significantly reduce NO production with respect to UT1, while T0 + T8 stimulation increases NO levels compared to UT1 (Figure 7B); UT2 and T0 + T24 show the same NO levels.

### 3.3. ECs Stimulation by RTP Promotes MMPs Expression and Cell Migration

An analysis of CMF capability to modulate the different steps of the angiogenic process was assessed. The first step of the angiogenic process requires a degradation of ECM by MMP activity: thus, the expression of MMP-2, MMP-9 and MMP-1 were evaluated. More precisely, a gene expression of *MMP-2* was evaluated 6 and 14 h after RTP stimulation, then after 14 and 24 h the protein expression of all three *MMPs* was assessed.

Six hours after 2 cycles and T0 + T8 RTP stimulation, a statistically significant increase in *MMP-2* gene expression is recorded, compared to UT1, while in the T0 + T24 stimulated cells *MMP-2* gene levels are comparable to UT2 (Figure 8A). Conversely, 14 h after treatment, *MMP-2* gene expression levels reached a plateau in 2 cycles and T0 + T8 RTP stimulation compared to UT1; although a slight increase is recorded in T0 + T24 stimulation, it is not statistically significant compared to UT2 (Figure 8B).

MMP-2 protein expression increases in a statistically significant manner after 14 h in 2 cycles and T0 + T8 conditions compared to UT1, reflecting the gene expression at 6 h, while its expression is significantly reduced after T0 + T24 stimulation compared to UT2 (Figure 9A). After 24 h, the significant increase in MMP-2 expression is still recorded in the condition of 2 cycles RTP compared to UT1; conversely, in the T0 + T8 condition MMP-2 levels reaches UT1 levels and in the T0 + T24 stimulation the level is comparable to UT2 (Figure 9B).

Western blotting analysis of MMP-9 reveals that after 14 h, in 2 cycles and T0 + T8 of RTP, the MMP-9 expression increases in a significant manner compared to UT1 (Figure 9C). The same trend is observed also for T0 + T24 stimulation that determines an increased MMP-9 expression compared to UT2. After 24 h, treated samples reach UT levels, except for T0 + T8 stimulation that maintains a statistically significant increase compared to UT1 (Figure 9D).

MMP-1 protein expression shows an opposite trend compared to both MMP-2 and MMP-9. After 14 h, only 2 consecutive RTP cycles increase in a significant manner MMP-1 levels compared to UT1, while the other conditions show MMP-1 levels comparable to not stimulated cells (Figure 9E). After 24 h the MMP-1 protein levels in 2 cycles and T0 + T8 stimulated samples further increase in a significant manner compared to UT1 (Figure 9F).

To understand if the increased expression of MMPs is paralleled by an enhancement of the basal EC migratory capability, a wound healing assay after RTP stimulation was performed (Figure 10). The cell monolayer scraped after 0 h from the stimulation represents the starting point (T0).

14 h after the stimulation, no statistically significant change in the cut width is recorded comparing stimulated and non-stimulated cells (UT). After 24 h, the highest and most significant reduction of cut width is recorded applying 2 consecutive cycles compared to UT1. A significant reduction is also evident after T0 + T8 and T0 + T24 RTP stimulation. After 48 h the same trend of 24 h is maintained (Figure 11).

### 3.4. CMF Stimulation Modulates Integrin β1-Erk-Cdk2 Pathway May Regulating Cell Growth Through G1/S Transition

The third angiogenic step after ECs migration is represented by cell proliferation. Thus, a possible molecular pathway, activated by RTP, controlling cell growth, driven by Integrin β1, Erk and Cdk2 proteins in EA.hy926 cells, was investigated.

After 14 h, 2 cycles and T0 + T8 stimulated samples show a similar Integrin β1 level with no statistically significant changes compared to UT1 (Figure 12A). Conversely, in T0 + T24 samples Integrin β1 expression with respect to UT2 is reduced. After 24 h, a statistically significant increase in Integrin β1 protein levels are recorded in 2 cycles and T0 + T8 conditions compared to UT1, while no changes are evidenced comparing T0 + T24 stimulation with UT2 (Figure 12B). At the same time, 14 h after stimulation with 2 cycles and T0 + T8 RTP there is a statistically significant decrease in p-Erk/Erk ratio compared to UT1, while a significant increase is recorded in T0 + T24 sample compared to UT2 (Figure 12C). On the contrary, 24 h after stimulation, p-Erk/Erk ratio increases in a statistically significant manner compared to UT1; in the T0 + T24 condition the increased levels recorded 14 h after stimulation, are maintained compared to UT2 (Figure 12D).

Cdk2 protein levels show a similar trend of Integrin β1: after 14 h, 2 cycles of stimulation induce a slight but statistically significant increase in Cdk2 expression compared to UT1, while T0 + T8 stimulation decreases Cdk2 level in a statistically significant manner respect to UT1 (Figure 12E). In addition, in T0 + T24 samples Cdk2 level significantly decreases compared to UT2. After 24 h, 2 cycles of stimulation keep significantly higher levels of Cdk2 compared to UT1, while T0 + T8 stimulation induces a reversed situation with respect to 14 h, with a statistically significant increase in Cdk2 level compared to UT1 (Figure 12F).

It is known that a regulation of the G1/S checkpoint is sustained by Integrin β1/Erk/Cdk2 pathway, and our results demonstrate a positive regulation of this pathway 24 h after 2 cycles and T0 + T8 stimulations. Thus, to understand if a consequent modulation of the G1/S checkpoint is recorded 38 h after CMF stimulation, a cell cycle analysis was performed by flow cytometry. A slight reduction of cell percentage in the G1 phase after 2 cycles and T0 + T8 stimulation is recorded compared to UT1, even if this reduction is not statistically significant. On the contrary, a significant increase in cell percentage in the S phase is evident and statistically significant after the aforementioned stimulations with respect to UT1 (Figure 13).

### 3.5. CMF Application Can Influence Endothelial Cells Morphogenic Process

The last angiogenic step is the morphogenic event, in which proliferated ECs rearrange each other to determine the tubule network formation. Thus, a tube formation assay was performed by using a Matrigel coating. The best tubule formation is appreciable after 5–10 h from the cell seeding on the Matrigel coating and already after 20–24 h there is a disruption of the formed network (Figure 14). For this reason, T0 + T8 and T0 + T24 represent two experimental points not suitable for this in vitro detection. On the contrary, 2consecutive cycles represent a good experimental point to understand if there is an in vitro effect on tubule formation induced by CMF application.

The densitometric analysis reveals that CMF application promotes in vitro tubules formation, with a statistically significant increase in mesh number (Figure 14A), segments (Figure 14B), and extremity (Figure 14C) in the 2 cycles of treated cells compared to UT1. Concerning the number of junctions, no significant changes are recorded between CMF application and UT1 (Figure 14D), while in the UT1, a higher number of nodes is appreciable (Figure 14E).

## 4. Discussion

Despite the growing evidence regarding the benefits of CMF application in different pathological conditions, such as muscular lesions, osteoarthritis, post-surgery regeneration, edema resorption, and many others, the biological and molecular events are poorly studied. Only a few publications are available concerning the in vitro effect of CMFs; the first one focused on the effect of CMFs in reducing the stressful stimulus of diabetic condition on fibroblasts and monocytes through a downregulation of Reactive Oxygen Species and pro-inflammatory cytokines levels (such as IL-1 and IL-6) with a consistent increase in anti-inflammatory cytokines (such as IL-10 and IL-12) [24]. More recently, the effect of CMFs on the immunity compartment was additionally studied by our group, pointing out the attention on Dental Pulp Stem Cells (DPSCs) and macrophages as the two main involved cellular types in dental regeneration. It emerged that CMFs could be an alternative therapy to the root canal treatment for dental caries, triggering the modulation of several macrophages’ polarization markers and superoxide anion levels, and, at the same time, promoting DPSC differentiation [29]. Considering the behavior of the vascular system in the healing processes of damaged tissues, the current study has the purpose of investigating the effect of CMFs on an in vitro model of ECs to understand the role of this new pulsed multifrequency electromagnetic fields’ application during the angiogenic process.

Although the use of cell lines can be recognized as a crucial study limitation, the in vitro model was established by using the EA.hy926 cell line, derived from the umbilical vein, due to its greater stability with respect to primary cells (e.g., HUVEC), and to the simple culture conditions not requiring additional culture components, such as growth factors.

Firstly, an investigation of CMF biocompatibility on the EC model was carried out: an RTP was applied to EA.hy926 cells, repeating the program for different cycles and at different time intervals in order to select the best RTP repetitions. The application of CMFs on ECs is biocompatible, independently from time intervals between RTP cycles: after 24 h, in most of the tested time points, an increased metabolic activity is observed. Indeed, it has been already demonstrated that MFs stimulate metabolic activity in different types of cells, such as osteogenic cells [30], muscle cells [31], and also in cancer cells in which, thanks to an effect on the metabolic process, a negative modulation of cell growth was recorded [32]. It is plausible that a similar effect is induced, in our model, by CMFs. The plateau recorded after 48 h of exposure could be due to the achievement of a confluence state, as also viable cells, counted by Trypan blue exclusion assay, demonstrate. In addition, the amount of released LDH after EC exposure to CMFs confirms the biocompatibility of the system.

Then, an in-depth analysis of CMF effect on the angiogenic steps was performed. As previously reported, angiogenesis is the process of new vascular network formation and it involves four primary steps: ECM degradation through MMP activity, cell migration, cell proliferation, and lastly, the morphogenetic event, in which cells migrated and proliferated in the area where new blood vessels are required, interacted with each other to obtain a new vascular network [5]. The first angiogenic step is represented by ECM degradation, coordinated by MMPs. This event is followed by the second phase which consists of ECs migration. These two stages are also crucial during the healing process due to the fact that damaged tissues promote the migration of ECs to speed up recovery events [33]. Therefore, it can be definitely stated that a system able to promote ECM degradation and ECs migration could be useful in the promotion of healing processes. MF application already demonstrated an appreciable effect in promoting ECM degradation and remodeling [34], nevertheless, no studies evidenced the effect of CMFs in this process. Our results show that CMF application, in particular 2 cycles and T0 + T8 stimulation, induces a modulation of *MMP-2* gene expression and MMP-1, MMP-2, and MMP-9 protein expression. The modulation of MMP-2 and MMP-9, whose role is mainly represented by ECM degradation to allow EC migration [22,35], is strictly in accordance with wound healing results, which evidenced in CMF-treated conditions, the highest cell migration starting from 24 h. This evidence lets us suppose a first ECM remodeling phase within 14 h after the stimulation, coordinated by MMP-2 and MMP-9, followed by cell migration induction. Conversely, a modulation of MMP-1 expression is highlighted after prolonged exposure let us suppose that the activity of MMP-1 is related to different cellular processes, such as cell proliferation rather than ECM remodeling. Indeed, it has been demonstrated by other authors that MMP-1 promotes Vascular Endothelial Growth Factor Receptor 2 (VEGFR2) expression and proliferation through PAR-1 and Nf-kB activation [36].

NO is a small signaling molecule synthesized by NOS enzymes: the inducible form, namely, iNOS, the neuronal one nNOS and the endothelial NOS [7]. NO plays a pivotal role in regulating different angiogenic events. In particular, it stimulates ECs migration through MMP-9 activation, regulates EC proliferation and survival and is essential in determining the final blood vessels shape. Post-translational modifications, such as phosphorylation and acetylation, are necessary to activate or inhibit eNOS activity; in particular, the main phosphorylation site, able to activate eNOS and induce NO production, is represented by ser1177, while the main inhibitory eNOS phosphorylation site is identified in thr495 [37,38]. CMF application, in particular T0 + T8 and T0 + T24 stimulations, seems to act through an activation of eNOS, responsible for NO production: an increased gene expression of *eNOS* is registered and, at the same time, an increased phosphorylation on ser1177, with a higher production of NO, is evident. Simultaneously, a decreased phosphorylation of thr495 is recorded. Although it is widely recognized that NO can control all the four phases of angiogenesis, positively regulating them [11], in our model, it seems to consistently modulate ECM degradation and cell migration being NO production and eNOS phosphorylation strictly in accordance with that recorded for MMP-9.

After ECM degradation and cell migration, the third angiogenic phase is ECs proliferation. Our data led us to suppose a boost of cell growth after CMF stimulation. In fact, from a morphological point of view, an increased number of cells in the treated conditions suggests a possible effect of RTP in stimulating cell proliferation. Again, available papers demonstrate the impact of MFs in promoting EC growth [22,39], but no scientific evidence is available regarding CMF effect. Even though the mechanism by which CMFs could promote cell growth still requires to be clarified, a possible pathway recruited to regulate EC proliferation could be Integrin β1-Erk-Cdk2 molecular cascade. Integrin β1 has a context-dependent function in ECs [12] as its activation can provoke Erk activation and its translocation into the nucleus resulting in a positive modulation of the cell cycle [13]. In fact, integrin-mediated cell adhesion is essential for G1/S checkpoint progression and the cyclin complex Cdk2/cyclin E appears indirectly modulated by Integrin-Erk activation, promoting G1 to S transition. 2 consecutive RTP cycles and T0 + T8 stimulation could enhance the aforementioned pathway; indeed, Integrin β1, p-Erk/Erk ratio, and Cdk2 show exactly the same increased trend. Moreover, a cell percentage reduction in the G1 phase and an increased percentage in S phase 14 h after the presumed activated Integrin β1/Erk/Cdk2 pathway led us to hypothesize a CMF control of the G1/S checkpoint, sustaining cell proliferation. Even if an increased cell number is already recorded 24 h after stimulation, it appears that a longer time after CMF exposure (24 h) could be required to involve and activate this signaling and sustain G1/S transition. It could be also argued that, before 24 h, alternative pathways are recruited to boost cell growth. Conversely, the T0 + T24 condition, showing different levels of Integrin β1/Erk/Cdk2 let the authors assume that a prolonged interval between two RTP cycles could recruit alternative mediators.

Along with cell proliferation, the morphogenic event, representing the fourth and culminating phase of angiogenesis, in which ECs reach a vessel conformation, occurs. The tube formation assay reveals that when 2 RTP cycles are applied some structures of the tubule network, such as meshes, segments, and extremity, are more abundant compared to non-stimulated cells, thus further confirming CMF capability to induce a more organized tubular network formation.

## 5. Conclusions

In conclusion, CMF application on ECs showed an appreciable ability in the modulation of the angiogenic events, with positive records in all steps required for new vessel formation, with a major extent to ECM remodeling, cell migration, and cell growth. Furthermore, these results let us hypothesize that CMF stimulation, in a damaged tissue, could exert a positive effect on the vascular system surrounding the injured areas, contributing to the promotion of tissue repair. It can be further argued that, due to the differences recorded in terms of pathway activations, the biological effect downstream of the activated pathway can be definitely influenced by the duration of treatment and by the time gap established between the cycles of treatment. Thus, it can be stated that the CMF device currently represents a well-founded non-invasive tool to be applied in various pathological conditions as a valid alternative to drug treatments or invasive interventions. However, further and future studies are certainly required to overcome possible limitations of the current study, such as (i) the obtained data referring to an in vitro model, commonly considered stable, controllable, and with most parameters known by the operator, possible discrepancies between in vitro and in vivo studies could occur; (ii) due to the simplicity of its setting up, the in vitro model represents a limited experimental approach, thus, the setting up of a more complex model (e.g., in vitro 3D models, spheroids, organoids) able to guarantee stability and, at the same time, more detailed information, is required; (iii) the current study reports data referring to ECs; however, other cell types, populating the connective compartment, are relevant to angiogenesis (e.g., fibroblasts), thus the CMF trial, in order to provide complete information regarding the effect of CMFs on angiogenic events, has to include further cell types. Furthermore, despite the encouraging clinical outcomes of CMF application, such as treatment of diabetic foot ulcers, hamstring trauma, burns and edema, until now little information on the biological processes is available. Thus, there is still a great deal of study to be conducted to elucidate the scientific basis of the reported clinical data, and our study represents one piece more in the investigation of the CMF effect.

## Figures and Tables

**Figure 1 cells-14-00332-f001:**
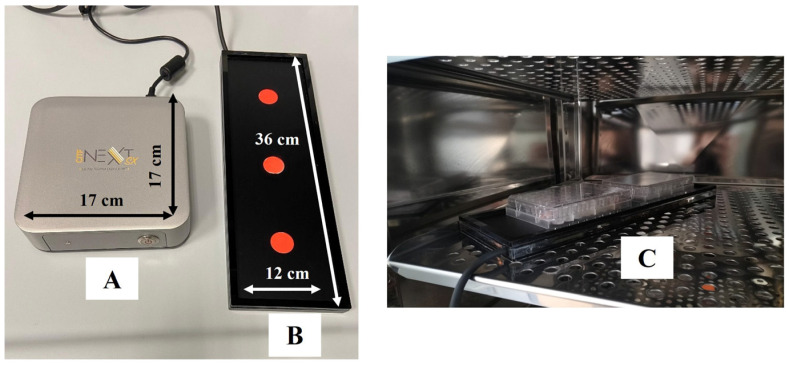
Next Generation CMFs instrument. The device is composed of the machine (**A**) and the CMFs emitting plaque (**B**). During the stimulation, the emitting plaque is located in the incubator with the multiwell plates on it (**C**).

**Figure 2 cells-14-00332-f002:**
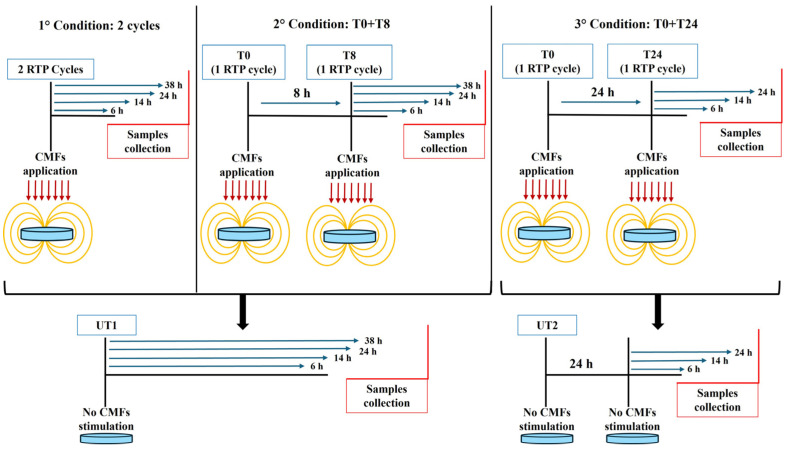
Study design of the selected experimental conditions.

**Figure 3 cells-14-00332-f003:**
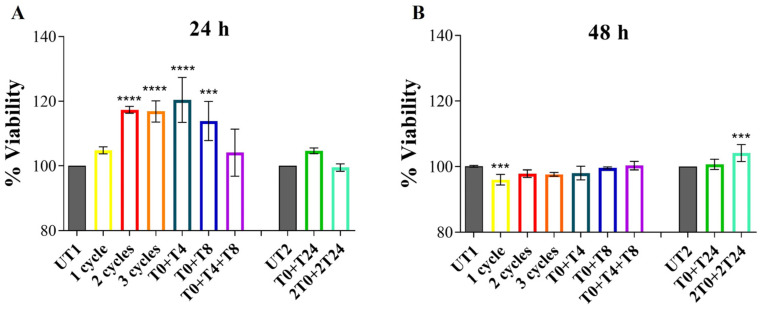
MTT test performed on EA.hy926 cell line 24 h (**A**) and 48 h (**B**) after stimulation with RTP of CMFs. **UT1** = cells with no RTP stimulation; **1 cycle** = cells stimulated with single RTP cycle; **2 cycles** = cells stimulated with two consecutive RTP cycles; **3 cycles** = cells stimulated with three consecutive RTP cycles; **T0 + T4** = cells stimulated with two RTP cycles with 4 h of interval from one to another; **T0 + T8** = cells stimulated with two RTP cycles with 8 h of interval from one to another; **T0 + T4 + T8** = cells stimulated with three RTP cycles with 4 h of interval from one to another; **UT2** = cells with no RTP stimulation but remain in culture one day more compared to UT1; **T0 + T24** = cells stimulated with two RTP cycles with 24 h of interval from one to another; **2T0 + 2T24** = cells stimulated with two consecutive RTP cycles with 24 h of interval. Metabolic activity of cells in the first treatment block has been normalized to UT1; metabolic activity of cells in the second treatment block has been normalized to UT2. Data shown represent the mean ± SD of three independent experiments. **** *p* < 0.0001; *** *p* < 0.001.

**Figure 4 cells-14-00332-f004:**
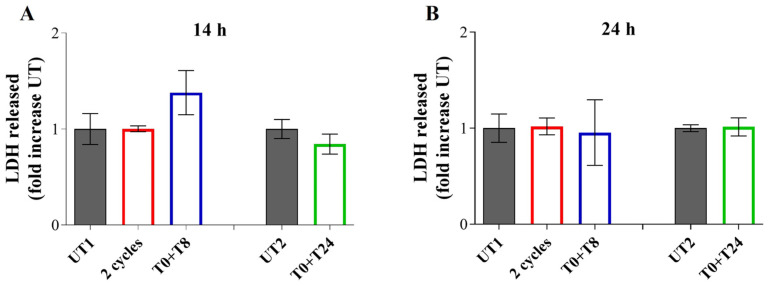
LDH release from EA.hy926 cells 14 h (**A**) and 24 h (**B**) after 2 cycles, T0 + T8 and T0 + T24 experimental conditions. Data shown represent the mean ± SD of three independent experiments.

**Figure 5 cells-14-00332-f005:**
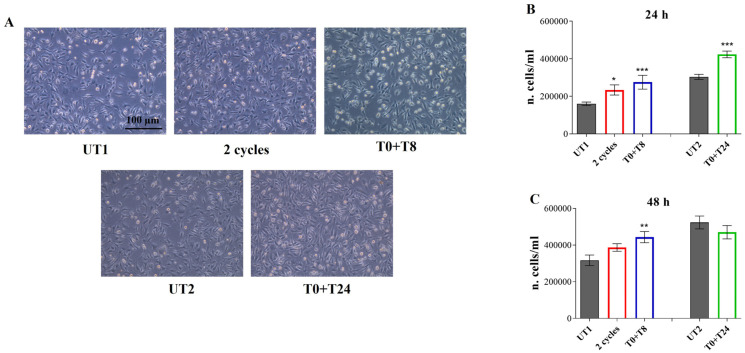
(**A**) Cell morphology observed by light phase contrast microscopy 24 h after 2 cycles, T0 + T8 and T0 + T24 RTP stimulations. Magnification, 10×. (**B**,**C**) Number of live cells/mL established by Trypan blue exclusion test 24 h (**B**) and 48 h (**C**) after 2 cycles, T0 + T8 and T0 + T24 RTP stimulations. Data shown represent the mean ± SD of three independent experiments. *** *p* < 0.001; ** *p* < 0.01; * *p* < 0.05. The number of cells in the first treatments block has been normalized to UT1; the number of cells in the second treatments block has been normalized to UT2.

**Figure 6 cells-14-00332-f006:**
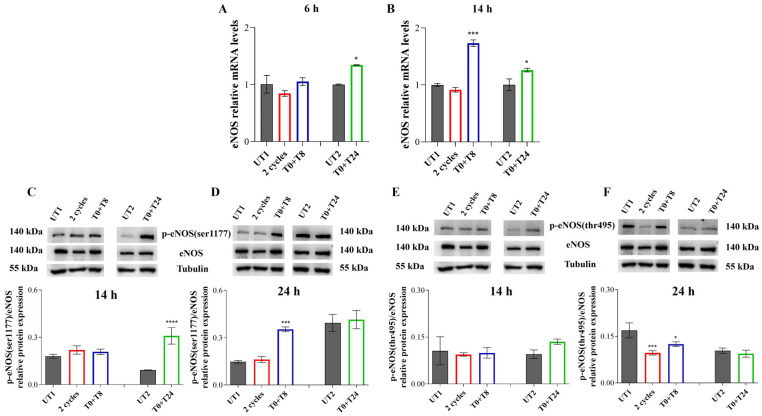
(**A**,**B**) Relative gene expression of *eNOS* in EA.hy926 cells 6 h (**A**) and 14 h (**B**) after stimulation with CMFs. The first treatments block data are expressed as relative to UT1 (calibrator sample, defined as 1); the second treatment block data are expressed as relative to UT2 (calibrator sample, defined as 1). Y-axis, fold change. Values represent means ± SD of three independent experiments. *** *p* < 0.001; * *p* < 0.05. (**C**,**D**) p-eNOS (ser1177), total eNOS immunoreactive bands and the p-eNOS (ser1177)/eNOS ratio relative protein expression levels in EA.hy926 cells 14 h (**C**) and 24 h (**D**) after stimulation with CMFs. (**E**,**F**) p-eNOS (thr495), total eNOS immunoreactive bands, and p-eNOS (thr495)/eNOS ratio relative protein expression levels in EA.hy926 cells 14 h (**E**) and 24 h (**F**) after stimulation with CMFs. The first treatments block data are expressed as relative to UT1; the second treatments block data are expressed as relative to UT2. Tubulin is used as a loading control. The bar graph displays densitometric values (reported as Integrated Optical Intensity) obtained from the immunoreactive bands analysis and are expressed as ±SD normalized on loading control. **** *p* < 0.0001; *** *p* < 0.001; * *p* < 0.05.

**Figure 7 cells-14-00332-f007:**
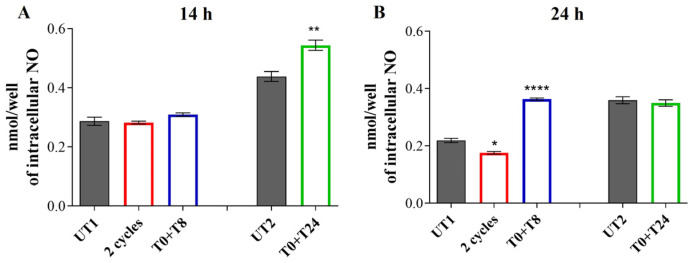
Intracellular NO levels 14 h (**A**) and 24 h (**B**) after CMF stimulation. The first treatments block data are expressed as relative to UT1; the second treatment block data are expressed as relative to UT2. Values represent means ± SD of three independent experiments. **** *p* < 0.0001; ** *p* < 0.01; * *p* < 0.05.

**Figure 8 cells-14-00332-f008:**
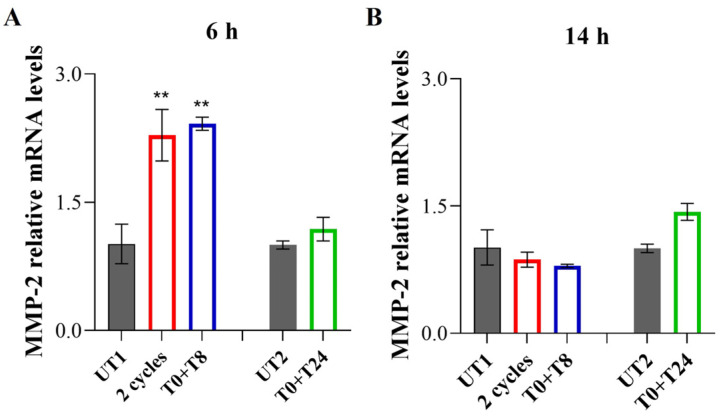
Relative gene expression of *MMP-2* in EA.hy926 cells 6 h (**A**) and 14 h (**B**) after stimulation with CMFs. The first treatments block data are expressed as relative to UT1 (calibrator sample, defined as 1); the second treatment block data are expressed as relative to UT2 (calibrator sample, defined as 1). Values represent means ± SD of three independent experiments. Y-axis, fold change. ** *p* < 0.01.

**Figure 9 cells-14-00332-f009:**
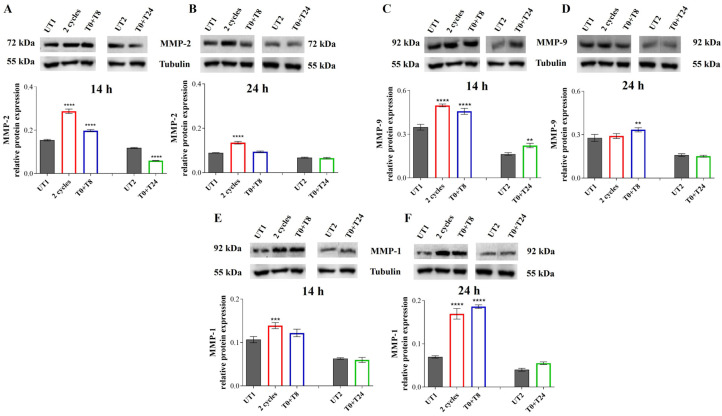
(**A**,**B**) MMP-2 immunoreactive bands and their relative protein expression levels in EA.hy926 cells 14 h (**A**) and 24 h (**B**) after stimulation with CMFs. (**C**,**D**) MMP-9 immunoreactive bands and their relative protein expression levels in EA.hy926 cells 14 h (**C**) and 24 h (**D**) after stimulation with CMFs. (**E**,**F**) MMP-1 immunoreactive bands and their relative protein expression levels in EA.hy926 cells 14 h (**E**) and 24 h (**F**) after stimulation with CMFs. The first treatments block data are expressed as relative to UT1; the second treatment block data are expressed as relative to UT2. Tubulin is used as a loading control. The bar graph displays densitometric values (reported as Integrated Optical Intensity) obtained from the immunoreactive bands analysis and are expressed as ± SD normalized on loading control. **** *p* < 0.0001; *** *p* < 0.001; ** *p* < 0.01.

**Figure 10 cells-14-00332-f010:**
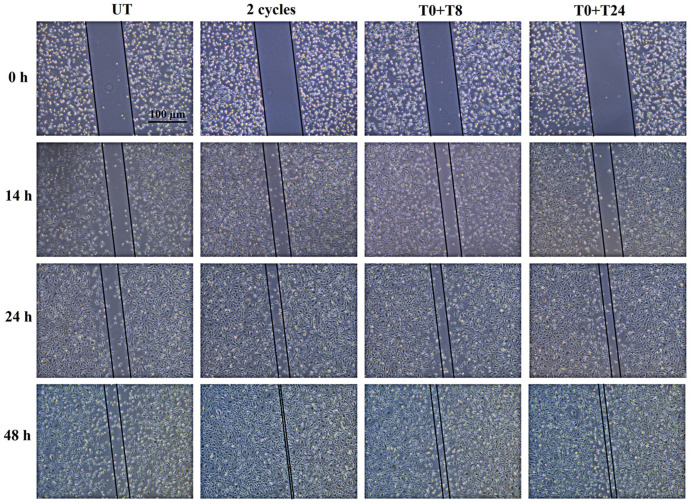
Representative images of scratch wound-healing assay performed on EA.hy926 cells at T0 (starting point, when cell monolayer was wounded) and 14, 24, and 48 h after 2 cycles, T0 + T8 and T0 + T24 RTP stimulations.

**Figure 11 cells-14-00332-f011:**
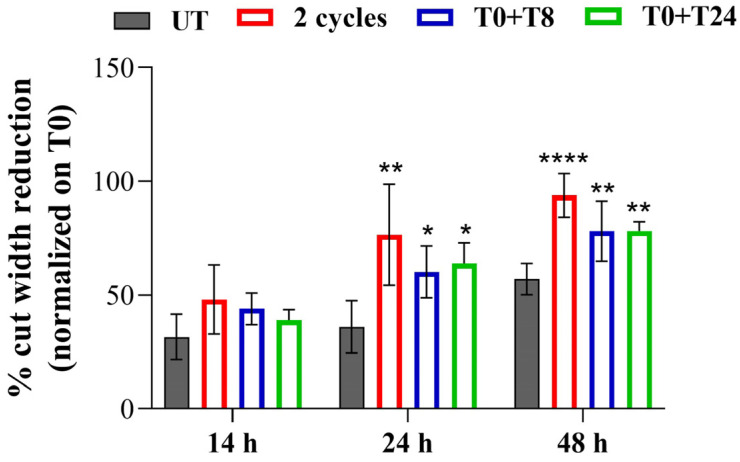
Bar graph displays the cut width reduction % obtained from scratch wound-healing assay, quantified by using ImageJ software. The cut reduction 14, 24 and 48 h after the stimulation is normalized on T0. **** *p* < 0.0001; ** *p* < 0.01; * *p* < 0.05.

**Figure 12 cells-14-00332-f012:**
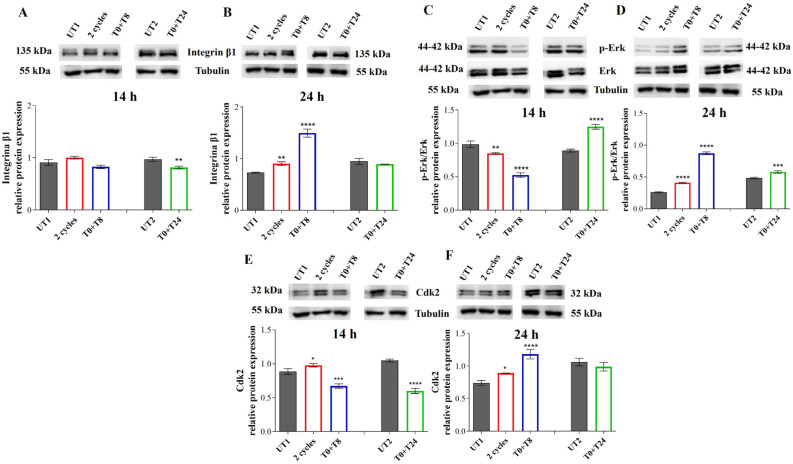
(**A**,**B**) Integrin β1 immunoreactive bands and their relative protein expression levels in EA.hy926 cells 14 h (**A**) and 24 h (**B**) after stimulation with CMFs. (**C**,**D**) p-Erk/Erk immunoreactive bands and p-Erk/Erk ratio relative protein expression levels in EA.hy926 cells 14 h (**C**) and 24 h (**D**) after stimulation with CMFs. (**E**,**F**) Cdk2 immunoreactive bands and their relative protein expression levels in EA.hy926 cells 14 h (**E**) and 24 h (**F**) after stimulation with CMFs. The first treatments block data are expressed as relative to UT1; the second treatment block data are expressed as relative to UT2. Tubulin is used as a loading control. The bar graph displays densitometric values (reported as Integrated Optical Intensity) obtained from the immunoreactive bands analysis and are expressed as ± SD normalized on loading control. **** *p* < 0.0001; *** *p* < 0.001; ** *p* < 0.01; * *p* < 0.05.

**Figure 13 cells-14-00332-f013:**
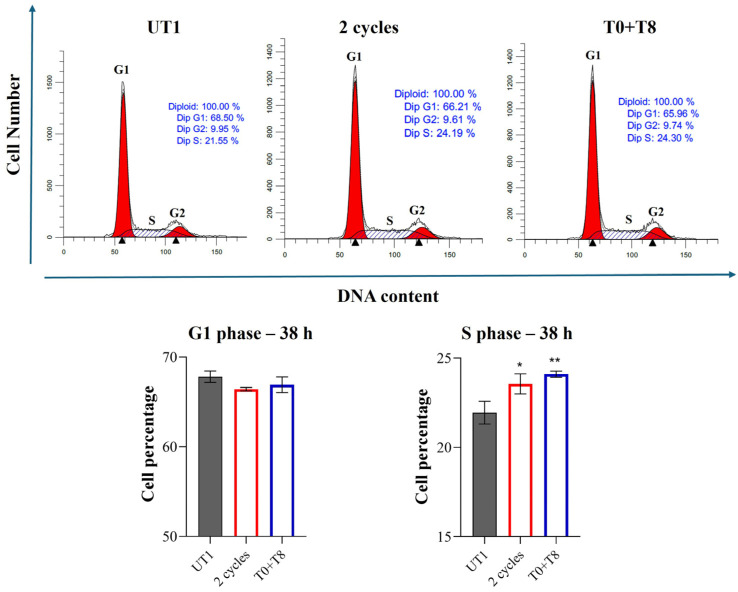
Cell cycle analysis in EA.hy926 cells 38 h after 2 cycles and T0 + T8 stimulation. The **upper panel** shows cell cycle profiles represented by fluorescence emission peaks obtained after PI staining (y-axis = cell count; x-axis = PI fluorescence emission in the FL-channel directly proportional with DNA content). The **lower panel** shows histograms reporting cell percentages in the G1 (left blue graph) and S (right red graph) phases of cell cycle. ** *p* < 0.01; * *p* < 0.05.

**Figure 14 cells-14-00332-f014:**
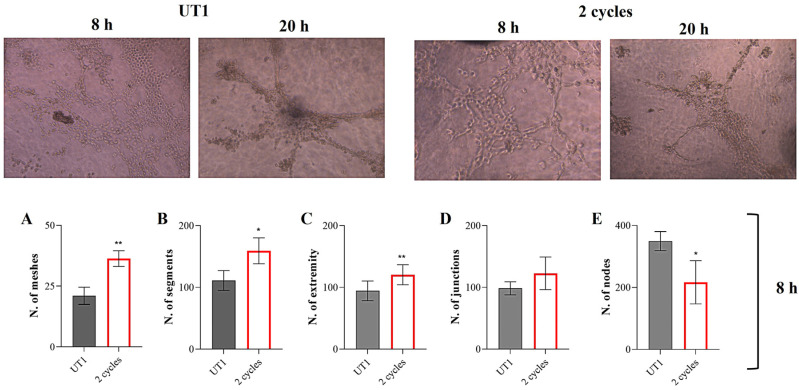
Tube formation assay upon CMF stimulation of EA.hy926 cells cultured on Matrigel-coated 96-well plate. (**Upper panel**) On the left: non-stimulated cells (UT1). On the right: stimulated cells with 2 RTP repeated cycles applied after 30 min from the cell seeding. The images were acquired after 8 h and 20 h from the CMF application. Magnification, 20×. One of the most representative images is shown for each experimental condition. (**Lower panel**) Densitometric analysis of tubules formation assay. From the images acquired after 8 h from CMF exposure, an analysis of meshes (**A**), segments (**B**), extremity (**C**), junctions (**D**), and nodes (**E**) numbers was carried out by using Angiogenesis Analyzer plugin (ImageJ). Values represent means ± SD of three independent experiments. Y-axis, fold change. ** *p* < 0.01; * *p* < 0.05.

**Table 1 cells-14-00332-t001:** CMF treatments applied by using RTP program.

**First Treatments Block: Short Interval Time Between Repeated RTP Cycles**	
UT1	Cells without RTP stimulation
1 cycle	Single RTP cycle
2 cycles	Two RTP consecutive cycles
3 cycles	Three RTP consecutive cycles
T0 + T4	One RTP cycle at T0 followed by an additional cycle after 4 h
T0 + T8	One RTP cycle at T0 followed by an additional cycle after 8 h
T0 + T4 + T8	One RTP cycle at T0, an additional cycle after 4 h, followed by a final cycle after 8 h from the T0
**Second** **Treatments** **Block: 24 h** **Interval** **Time** **Between** **Repeated RTP** **Cycles**	
UT2	Cells without RTP stimulation kept in culture one day more compared to UT1
T0 + T24	One RTP cycle at T0 followed by an additional cycle after 24 h
2T0 + 2T24	Two RTP cycle at T0 followed by an additional two cycles after 24 h

**Table 2 cells-14-00332-t002:** Primer sequences for quantitative PCR.

Gene	Sequence (5′–3′)	Reference
*GAPDH-FW*	GGGTGTGAACCATGAGAAGTA	Primer blast
*GAPDH-RW*	ACTGTGGTCATGAGTCCTTC	Primer blast
*eNOS-FW*	GTGTCCCTCGAACACGAGAC	Primer blast
*eNOS-RW*	AGTGGGTCTGAGCAGGAGAT	Primer blast
*MMP-2-FW*	GCTACGATGGAGGCGCTAAT	Primer blast
*MMP-2-RW*	GGGCAGCCATAGAAGGTGTT	Primer blast

## Data Availability

The original contributions presented in this study are included in the article. Further inquiries can be directed to the corresponding author.

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
