# Peer review of "Angiogenic Events Positively Modulated by Complex Magnetic Fields in an In Vitro Endothelial Cell Model"

_cells, 2025, doi:10.3390/cells14050332_

Round 1

Reviewer 1 Report

Comments and Suggestions for Authors

The work of Ricci et al. is an interesting manuscript that deals with the effects of (complex) magnetic fields on cells, in this case endothelial cells. Basically, the manuscript is written in a good scientific style and addresses interesting effects on viability, nitric oxide metabolism and the expression of selected endothelium-specific enzymes (MMPs), which also offer starting points for therapeutic applications. Regardless, from the reviewer's point of view, there are some major concerns that need to be addressed. In its current form, the manuscript cannot be recommended for publication.

Major concerns

The number of the conducted experiments is a crucial point and must be addressed.

Experimental setup: A more thorough description of the experimental setup is necessary to facilitate comprehensional experiments (Figure 1). The dimensions of the machine and the CMF-emitting plate are not clearly specified. An inclusion of a scale bar or cm values would enhance clarity. Additionally, the magnetic field generated should be illustrated to facilitate comprehension of its location and direction. It is imperative to ascertain whether multiple fields are present or if it is just one large field. Also, it should be clarified how many well plates can be placed on the CMF emitting plate at once. To facilitate understanding, it is recommended to either create an illustrative representation of the entire apparatus or to take a photograph under the conditions in the incubator.

Additionally, the comparison of exposed cells with UT1/UT2 in the experimental setup shown in Figure 2 should be explained.

Western blots: The figure captions (Figures 6, 9 and 12) should include a more detailed explanation of the Western blot images, as text alone is not sufficient to understand the figures. In addition, it is imperative to explain the temporal context in which the Western blots were performed, as well as their relationship to the graphs, to ensure consistency. To address these concerns, it is recommended to create a figure that integrates the Western blot into the overall design of the figure, thus facilitating its connection to a specific time point

Definition of controls: It is not advisable to compare the selected time points with the applied controls. The application of UT1 is limited to a single comparison of the magnetic field application, since the cells undergo a steady proliferation. A true control condition should be defined by identical time (!), growth conditions and environment to that of the treatment condition. A valid approach would be to compare UT1 and 2 RTP cycles, provided that both cells are analyzed after 6 hours, together with the duration of both RTP cycles (+2 x 28 min / ~ 1h). In this case, the analysis time for both points is more closely related and can be neglected if necessary. However, for T0 + T8, the proliferation time of 8 hours for the control is also considered as “experimental time”, giving a total of 8 + 6 hours. Therefore, to ensure the accuracy and reliability of the results, it is recommended to extend the experiment with additional controls that have exactly the same conditions. This approach facilitates the establishment of more precise controls and allows the formulation of a reliable statement about the observed effects. The lack of consistency in the controls constitutes a problem in all subsequent experiments, as the designated time points are inaccurate. For the MTT (2.3), the medium was removed after 24 and 48 hours; however, this time is not consistent for all RTP cycles (+28 minutes per cycle). Consequently, the control must be analyzed after 24h + 4h (+(+2 x 28 min / ~ 1h)), yet it cannot be compared to UT1, which was measured after 24h only. Due to the continuous proliferation of cells, the number of cells after 4 hours (2 x 28 min / ~ 1h) + 24 hours is a while longer than the number of cells after 24 hours only. A similar observation can be made in the case of Trypan blue (2.4), as the measurement was conducted following the CMF stimulation. Consequently, a direct comparison between T0+T8 and UT1 (8h+24h) is not possible. In this case, it is recommended to perform a protein determination to normalize the data based on the amount of protein at each time point. In addition, it is imperative to improve the accuracy of UT2. It is also imperative to specify the reason why the cells are " one day more in culture than UT1". To ensure the validity of the results, it is important to apply the same conditions to both the untreated cells (UT) and the treated cells for both cycles (24h + ~1h) and the 6-hour control period.

Wound healing assay: The initial images captured at 0 hours reveal discrepancies in the dimensions of the scratches. It is necessary to ascertain the total number of experiments conducted to obtain a comprehensive dataset. The the mean size of the scratches has to be ascertained. Given the observed variation in scratch size, methodologies from companies such as ibidi (see https://ibidi.com/culture-inserts/24-culture-insert-2-well.html?gad_source=1&gclid=EAIaIQobChMIjq-hiMf8igMVbDIIBR2FDyH0EAAYASABEgKIU_D_BwE) or Sartorius may be employed to ensure more uniform scratches with reduced size variability and besides that, are not that “harmful” compared to the use of a pipette tip. Further, a notable variation in cell number is evident between UT2 and T0+T24, which contributes to the previous observations. 

Minor points

There are some inconsistencies in the writing style, such as the use of “EA-hy926” and “EA.hy926”, “microT” and “µT”, and “cicli” and “cycle”, which need to be corrected and standardized throughout the entire manuscript.

In addition, the formatting needs to be checked, especially the spacing (spaces) between units and values. In particular, use a non-breaking (fixed) spaces between values and units.

The cell seeding number and the type of multi-well plates must be specified to ensure the reproducibility of the experiments.

In addition, a short description and explanation of the selected cell line should be included to understand why it was chosen.

The description of the NOS metabolism (line 504 and following) and the integrin-β1 signaling pathway (line 534 and following) should be moved to the introduction as prior information.

For the entire document, it would be desirable to use a consistent set of colors and formatting for the figures (e.g. control is always black, same exposure settings always have the same color).

Of note, the significance levels vary considerably between results. For significance levels that currently vary from result to result (** p < 0.002, ** p < 0.0047, ** p < 0.01, etc.), a consistent type should be chosen, *p<0.05, **p<0.01, ***p<0.001). Are the data normally distributed so that the use of parametric tests is justified? Do the repetitions carried out in x independent experiments justify the indication of highly or particularly highly significant levels? Please consider also the major concern on use/definition of controls.

In Figure 5, the lack of a scale is noticeable. For the wound healing test (see also major concerns), the manufacturer of the pipette tips should definitely be added.

In Figure 13, please use G1 phase or S phase as headings for the lower graphs.

In Figure 14, the description of the upper panel (left/right) should be corrected.

Reviewer 2 Report

Comments and Suggestions for Authors

I wish to express my appreciation for the quality of your work. The methodological rigor, clarity in the presentation of results, and depth of discussion are evident. Your ability to identify a knowledge gap in the field and design an experiment to address it is commendable.

I consider your results on the biocompatibility of CMFs and their effect on the metabolic activity of endothelial cells to be very promising. However, I suggest delving deeper into the following aspects:

a) Molecular mechanisms: It would be interesting to explore the specific signaling pathways that could be involved in the cellular response to CMFs. Are there previous studies that suggest possible molecular targets?

b) Comparison with in vivo: How do your in vitro findings relate to previous vivo studies? What factors could explain possible discrepancies?

c) Other cell types: Have you considered evaluating the effect of CMFs on other cell types relevant to angiogenesis?

d) Clinical implications: How could your findings be translated into clinical applications, such as promoting wound healing?

e) Limitations of the model: What are the inherent limitations of an in vitro model, and how could these be addressed in future studies?

Reviewer 3 Report

Comments and Suggestions for Authors

Ricchi and colleagues report the effects of the composite magnetic field (CMF)  regimens used in clinical applications upon the characteristics of an endothelial cell (EC) culture. They found that in EC culture CMF increases proliferation and migration, production of several metalloproteinases, expression of Integrin beta 1 and cyclin dependent kinase 2, and activates phosphorylation of Erk and eNOS. They suggest that the healing effects of CMF observed in vivo could be at least partially explained by the stimulation of EC which results in enhanced angiogenesis.

Critiques:

1.       What is the species origin of EC used in this paper? Are they arterial, venous or microvascular? Based on growth in DMEM with serum but without addition of growth factors, those might be rodent EC. Why not to use better characterized diploid cultures of human EC?

2.       The quality of phase contrast photos is very low, especially when tube formation is presented. Also, color is out of place in phase contrast images.

3.       Mentioning integrin beta1 – Erk -CDK2 signaling pathway as a potential culprit in CMF effects is a speculation. The observed changes, especially enhancement of integrin expression, may be not connected to each other.

4.       In addition to CDK2, the authors should also explore the expression of cyclins D and A2, which is much more dynamic and better represents the proliferative status of the cells.

Round 2

Reviewer 3 Report

Comments and Suggestions for Authors

The authors properly answered the critiques

Author Response

The authors thank the referee for his positive evaluation.